# Seabirds as samplers of the marine environment – a case study in Northern Gannets

Stefan Garthe[1], Verena Peschko[1], Ulrike Kubetzki[1,2], Anna-Marie Corman[1]

[1]Research & Technology Centre (FTZ), Kiel University, Hafentörn 1, D-25761 Büsum, Germany
[2]Department of Animal Ecology and Conservation, Biocentre Grindel, Hamburg University, Martin-Luther-King Platz 3, 20146 Hamburg, Germany

*Correspondence to*: Stefan Garthe (garthe@ftz-west.uni-kiel.de)

**Abstract.** Understanding distribution patterns, activities, and foraging behaviours of seabirds requires interdisciplinary approaches. In this paper, we provide examples of the data and analytical procedures from a new study in the German Bight,
North Sea, tracking Northern Gannets (Morus bassanus) at their breeding colony on the island of Helgoland. Individual adult Northern Gannets were equipped with different types of data loggers for several weeks, measuring geographic positions and other parameters mostly at 3-5 min intervals. Birds flew in all directions from the island to search for food, but most flights targeted areas to the (N)NW of Helgoland. Foraging trips were remarkably variable in duration and distance; most trips lasted 1–15 h and extended from 3–80 km from the breeding colony on Helgoland. Dives of gannets were generally shallow, with
more than half of the dives only reaching depths of 1–3 m. The maximum dive depth was 11.4 m. Gannets showed a clear diurnal rhythm in their diving activity, with dives being almost completely restricted to the daylight period. Most flight activity at sea occurred at an altitude between the sea surface and 40 m. Gannets mostly stayed away from the wind farms and passed around them much more frequently than flying through them. Detailed information on individual animals may provide important insights into processes that are not detectable at a community level.

## 1 Introduction

Seabirds are marine animals that live mostly at or near the air–water interface. The dynamics of both these media may consequently have a strong influence on the ecology of seabirds (Schneider 1991). Many studies in the world's oceans have shown that the physical environment has a substantial influence on seabird distributions (e.g. Briggs et al., 1987, Hunt, 1990). Physical processes are particularly relevant to seabirds when they cause predictable prey aggregations, either regular or
irregular. However, other opportunities (such as fisheries discards; e.g. Ryan & Moloney, 1988, Garthe et al., 1996) and constraints (such as the need to breed on land; e.g. Schneider & Hunt, 1984, Wilson et al., 1995a) may also influence seabird distributions and related behaviours. An essential feature in the marine system is 'scale'. Quantitative relations between abiotic and biotic variables are strongly influenced by the scale at which they are measured (Schneider, 1994). Thus, general seabird distribution patterns often correspond best with physical phenomena at large scales, whereas smaller scale patterns are
associated with biological features such as foraging range, social interactions, and prey availability (Schneider & Duffy, 1985,

Hunt & Schneider, 1987). Most behaviours at sea are directly related to foraging (e.g. searching, feeding) or the result of foraging-related constraints (waiting for food to become available, digesting, commuting). Several external and internal characteristics and limitations influence foraging activities, e.g. diurnal rhythms, flight manoeuvrability, feeding techniques, prey-detection capabilities, social attractions, learning and age-dependent skills, foraging ranges, and dietary preferences (Furness & Monaghan, 1987, Shealer, 2002).

For decades, studies of seabird biology were mainly land-based, with a particular focus on the breeding period. Although there were understandable logistic reasons for this, it has led to severe biases in our understanding of seabird ecology. Two subsequent approaches focusing on the behaviours of seabirds at sea have allowed significant progress in our understanding of seabird ecology. One such approach involved studying seabird distributions at sea from boats. Whereas early work was targeted towards establishing the distribution patterns of seabirds (e.g. Brown, 1986, Tasker et al., 1987), later studies concentrated on improving our understanding of the underlying factors, including habitat parameters, mainly hydrographic features measured synoptically at sea or by remote techniques, and food availability, assessed by detecting and possibly quantifying prey at sea (e.g. Hunt et al., 1998, Davoren et al., 2003, Jahncke et al., 2005). The second approach was to equip seabirds with telemetric devices and/or data-logging units (e.g. Jouventin & Weimerskirch, 1990, Wilson et al., 2002, Wilson & Vandenabeele, 2012). These devices record the bird's position and/or other parameters such as temperature and depth while the bird is at sea. Because seabirds are fast-moving and wide-ranging animals, this approach also enables us to study them in logistically inaccessible areas. Furthermore, it allows information on individual birds to be collected, in contrast to boat-based observations, which involve larger samples of birds, but where single individuals cannot be tracked over larger areas or time spans.

Understanding patterns in distributions, activities, and foraging behaviours of seabirds requires interdisciplinary approaches. The physical properties of the sea establish the basic habitat parameters with which both the seabirds and their prey have to cope, while biological conditions influence the birds' food supply (e.g. by prey behaviour) and foraging behaviours. Furthermore, anthropogenic activities may also influence different aspects of the marine environment, both directly on individual seabirds, and indirectly by affecting habitat conditions and prey availability. A combination of these methodological and conceptual approaches will further improve our understanding of the ecology of seabirds within the study area.

In this paper, we provide an overview of a new study in the German Bight, North Sea, connected to the Coastal Observing System for Northern and Arctic Seas (COSYNA) network. We started tracking Northern Gannets (*Morus bassanus*) at their breeding colony on the island of Helgoland in 2014 (Garthe et al., 2016). Gannets were selected as they have the largest foraging ranges of all abundant seabird species on Helgoland and are large animals that can carry various types of data loggers. Here we provide examples of the data and analytical procedures based on selected data sets from 2015, and explain the value and perspectives of such studies, especially in relation to coastal observation systems such as COSYNA.

## 2 Methods

### 2.1 Field work

Field work was conducted on the island of Helgoland (54° 11' N, 7° 55' E) in the southeastern North Sea (Fig. 1). A total of 14 adult Northern Gannets that were either incubating or rearing chicks were caught on 12–13 May, 17–18 June, or 22–23 July

2015. All birds were equipped with data loggers. Ten gannets each received a Bird Solar GPS logger (e-obs GmbH, Munich, Germany) and the other four birds were equipped with both a CatLog-S GPS logger (Catnip Technologies, Hong Kong, China) and a precision temperature–depth (PTD) logger (Earth & Ocean Technologies, Kiel, Germany). All loggers were attached to the base of the four central tail feathers using TESA® tape (Beiersdorf AG GmbH, Hamburg, Germany; Fig. 2). Data obtained from these loggers covered durations of 0.4–10.9 weeks.

The total masses of the attached devices (including sealing, base plate, and tape) were about 48 g (Bird Solar) and 64 g (CatLog-S plus PTD), representing 1.5% and 1.9%, respectively, of the mean gannet body mass of 3,286 g (Wanless & Okill, 1994). This is well below the potential threshold of 3% (Phillips et al., 2003, but see Vandenabeele et al., 2012). Although attachments to the tail may have a negative influence on flight behaviour (Vandenabeele et al., 2014), most pairs successfully incubated their eggs and/or raised their chicks, similar to non-handled nests, with no visible effects on bird behaviour.

### 2.2 Technology

#### 2.2.1 Bird Solar GPS logger

These loggers recorded date, time, position (latitude, longitude), ground speed, heading and acceleration. The sampling interval was mostly set to 3–5 min, and the triaxial accelerometer to 0.25–3 min. The onboard memories were either 32 or 64 MB. The outer diameters of the devices were $63 \times 22 \times 16$ mm, plus base plates and an antenna of 76 mm for data transfer. Data could

be downloaded remotely using a hand-held device when approaching the birds in the colony.

#### 2.2.2 CatLog-S GPS logger

These devices recorded date, time, and position (latitude, longitude) and were set at an interval of 5 min. Dimensions varied slightly according to battery type, but were about $50 \times 35 \times 8$ mm. The plate was encased by a heatable plastic housing. Data were retrieved by recapturing the bird and downloading data from the device.

#### 2.2.3 PTD loggers

These loggers had 2 MB onboard memory and measured date, time, pressure, and internal and external temperatures (Earth & Ocean Technologies). Temperature measurements were obtained from an external, fast-responding, temperature sensor that allowed sampling of the water column with minimal time lag in thermal signals (temperature-response time T 0.9 (i.e. time to reach 90% ΔT, following a temperature change) of ca. 1.8 s (Daunt et al., 2003). The streamlined lightweight carbon-fibre–

composite casing (outer diameter 19 mm, length 80 mm) weighed about 23 g. Recording intervals for temperature and pressure were set at 3 s. Data could be retrieved by recapturing the bird and downloading the data from the device.

## 2.3 Northern Gannets

The Northern Gannet is the largest seabird species in the North Atlantic. It has a body mass of 2.3–3.6 kg and breeds in colonies of up to several tens of thousands of pairs. Northern Gannets spend their entire life at sea, except for breeding on land. They usually start breeding at 5–6 years old, and may live to 20 years or older. They lay one egg that is incubated for 6 weeks, followed by a chick-rearing period of about 13 weeks. Only one adult of the pair is usually at the nest at any one time during incubation or chick guarding, while the other is at sea (Nelson, 2002, Bauer et al., 2005). Apart from short flights to collect nesting material or due to disturbance/interactions at the nest site, gannets carry out foraging trips to collect food for themselves and their offspring. They usually forage using so-called plunge dives, which are initiated when flying (and searching) at a few to several tens of metres above the sea surface (Nelson, 2002, Garthe et al., 2014). Two different dive types can be distinguished in this species (Garthe et al., 2000). U-shaped dives occur when the birds remain at a largely constant depth for a period after plunging into the water, with little vertical movement, before returning to the sea surface. In contrast, V-shaped dives are usually short and shallow, with the ascent almost immediately following the descent.

Northern Gannets have recently been studied intensively by satellite telemetry and data loggers in various places (e.g. Hamer et al. 2001, Pettex et al. 2012, Wakefield et al. 2013), thus allowing comparisons among regions and populations.

## 3 Products and analyses

### 3.1 Flight patterns

Figures 3a and 3b show the flight patterns of two adult Northern Gannets that were typical of 13 of the 14 individuals tracked in 2015. Birds flew in all directions from the island to search for food, but most flights targeted areas to the (N)NW of Helgoland. Foraging trips (defined in this paper as absences from the nest site of at least 20 min and of at least 2.0 km direct distance) were remarkably variable in duration and distance; most trips lasted 1–15 h and extended from 3–80 km from the breeding colony on Helgoland. One individual's behaviour differed from that of the other gannets by repeatedly flying far north to forage in the Skagerrak (Fig. 3c). These long-distance foraging trips were almost identical in their structures (n = 3; duration = 44.3–59.3 h, most distant location = 375–388 km, total distance flown = 971–1019 km) and were interspersed with 'normal' foraging trips into the German Bight.

### 3.2 Diving behaviour

Dives of gannets breeding on Helgoland and foraging in the (south-)eastern North Sea were generally shallow, with more than half of the dives only reaching depths of 1–3 m (Fig. 4). The maximum dive depth was 11.4 m, and the median dive depth was 2.2 m (n = 4 individuals, n = 2,577 dives). Most dives were V-shaped, though the measuring interval of 3 s did not allow a

precise determination of the proportions of U- and V-shaped dives (see Garthe et al., 2000). The measuring interval of 3 s might also mask the deepest parts of some dives and may thus underestimate dive depth in general. We therefore re-analysed a random sample of 100 dives from Garthe et al. (2014), where gannets exhibited a similar high percentage of V-shaped dives, using both 1 s and 3 s intervals (10 individuals, 10 dives each). Scaling down to 3 s missed 10% of the dives, while the median detected dive depth was only slightly smaller (4.3 vs. 4.5 m). These subtle differences demonstrate the validity of 3 s measuring intervals to determine the dive-depth pattern.

These dives were shallow compared with previous studies conducted in the northwestern North Sea (Lewis et al., 2002), the English Channel (Grémillet et al., 2006), the northwest Atlantic (Garthe et al., 2000), and the Gulf of St. Lawrence, Canada (Garthe et al., 2007). Although the sample sizes of individuals is small, this does not hold true for the number of days the birds were tagged and the number of dives. Dive depths recorded from Helgoland gannets in 2015 remained much shallower even when subsampling small data sets from a large data base from eastern Canada (Garthe et al., unpubl. data).

Gannets showed a clear diurnal rhythm in their diving activity (Fig. 5). Dives were almost completely restricted to the daylight period, with the remaining dives occurring around dawn and dusk. No dives were recorded between 22:09 and 05:21 Central European Summer Time (CEST). This pattern fits well previous studies (Garthe et al., 2000, Garthe et al., 2003).

### 3.3 Habitat analyses

Dive positions were analysed for the fixed habitat parameters distance to colony (Helgoland), water depth and distance to nearest land (except for Helgoland; Fig. 6). Almost two thirds of the dives were carried out at a distance of less than 50 km from the colony, with proportions declining further away from the colony. However, at the largest distances, proportions of dives increased again, strongly indicating that gannets may have specifically targeted such distant foraging areas (see also chapter 3.5, Fig. 9). As related to water depth, gannets from Helgoland were diving most often in waters of 20-40 m depths; less often in shallower, and rarely in deeper waters (Fig. 6). For foraging, gannets mostly stayed away from the coast, with highest proportions at a distance of 40-60 km. This pattern differs completely from studies in eastern Canada where gannets were found to concentrate their diving efforts on the coastal zone (Garthe et al., 2007). Both Helgoland (located ca. 43 km north of the East Frisian islands) and Funk Island (Newfoundland, Canada; located ca. 50 km away from the coast) have a similar placement and, in consequence, the location of the colonies cannot explain the observed difference in coastal focus. However, the near-coastal waters in the Canadian study sites are characterised by much more marine conditions and larger water depths compared to the Wadden Sea coast with extended shallow waters in the German Bight.

A Gaussian linear mixed model was used to analyse the impact of the habitat parameters distance to colony, water depth and distance to nearest land on 1) the dive depth, and 2) the dive duration of the tagged birds (using R version 3.3.2, R Development Core Team, 2016; package 'lme4' by Bates et al., 2015). Both response variables were log-transformed to approach normality. The three habitat parameters were used as numeric explanatory variables. Dive ID nested within bird ID was taken as random factor to avoid pseudo replication due to multiple measurements per bird. Model selection was performed using the function 'drop1' to find the relevant habitat parameter explaining the variance of dive depth and/or dive duration. This

function tests every term in the model as if it was the last entering the model. In turn, every term in the model is omitted, and the reduced model is then compared to the full model by a likelihood ratio test under 1 degree of freedom (Korner-Nievergelt et al. 2015). Though the sample size of individuals was low, the temporal coverage (12-18 days) and the number of dives per individual (381-773) were high. In this data set dive depth could not be explained statistically by any of the three fixed habitat variables while dive duration could be explained by water depth (Table 1). It is to be expected that larger data sets that will be collected in the future may exhibit more significant relationships to these and other habitat parameters.

## 3.4 Flight altitudes

The altitudes of flying birds are important in relation to their migratory movements, prey-searching behaviour, and potential overlap with technical installations at sea.

The BirdSolar GPS loggers calculated height above the ellipsoid when connecting with satellites during positional fixes, and altitude measurements were therefore corrected for geoid height (39.1 m at colony location). Altitude estimates are improved when connection time to the satellite is increased (e.g. Corman & Garthe, 2014), and we therefore used pulses of GPS fixes over 11–15 s and analysed the last and assumed best altitude measurement from each pulse. Figure 7 shows non-smoothed altitude measurements for one foraging trip of 22.7 h. Colony attendance was derived from positional fixes and known nest position, on-water periods were determined from ground speed ($< 3$ km h$^{-1}$) and positional fixes. Although values fluctuated slightly even for fixed places such as the nest site in the colony and the sea surface, the measurements appeared reasonable and showed that most flight activity occurred at an altitude between the sea surface and 40 m, with maximum values in this study when birds were commuting to/from the colony. Measurements in the colony and on water can be used to calibrate altitude measurements because of their relatively well-known heights.

Flight heights of gannets have also been determined by radar measurements (e.g. Krijgsveld et al., 2011), visual observations (e.g. Johnston et al., 2014), and from pressure sensors (Garthe et al., 2014, Cleasby et al., 2015). Overall, flight heights of gannets tend to be low, with relatively few flights above 50 m and very few recorded above 100 m, though no comprehensive analysis has yet been published.

## 3.5 Behavioural patterns

Animal movements can be tracked using motions sensors. Many data loggers contain accelerometers that ideally cover all three axes (x, y, z), and frequent measurements allow behavioural differentiation at a fine scale (e.g. Sakamoto et al., 2009).

Figure 8 shows an example of accelerometer measurements of a Northern Gannet at a 0.25 min interval over 24 h. Recordings start when the bird is on its nest, with very little activity, obviously sleeping. After about 3 h, the bird remains on its nest but its activity increases, coinciding with dawn. A few hours later, the bird leaves the nest and flies off, followed by a period of about 10 h of mostly flying, interrupted by a few shorter swimming periods. Towards the end of the recording period, the bird settles down on the sea surface and remains floating there overnight (Fig. 8). Such information is important in many ways. It may help identifying the relevance of certain sea areas; i.e. whether areas are used for foraging, or just for resting, or

for long(er)-distance movements. Quantifiying birds' activities is a well-established tool to measure energy expenditure. Such energy budgets may e.g. help unraveling seabird movement strategies as has been shown by Garthe et al. (2012) for Northern Gannets wintering in different regions of the Northeast Atlantic.

To determine the relevance of certain sea areas and to improve our understanding of the flight patterns of the birds, it is necessary to know when and where the birds are feeding. Because gannets almost always obtain food by plunge diving, observing dives provides a good proxy for determining feeding areas. Figure 9 shows the flight tracks and dive locations of the gannet that flew repeatedly towards the Skagerrak. It shows that the gannet was foraging intensively in the Skagerrak, while longer passages on outbound and inbound flights were long-distance flights without much foraging activity.

## 3.6 Overlap with human pressures

The ability to track seabird movements at small spatial and temporal scales makes it possible to study the potential impacts of human activities at sea comprehensively. Twelve offshore wind farms have been built and became operational in the German Bight between 2008 and November 2016, and a further five are currently under construction. Another 15 wind farms have been given consent, and several tens more have been applied for. The impact of wind farms on seabirds, which is currently a hot topic in conservation biology and environmental policy (e.g. Furness et al., 2013, Masden et al., 2015), can thus be studied comprehensively in German North Sea waters. In 2014, gannets were tracked near existing wind farms for the first time. All three individuals largely avoided the wind farm area north of Helgoland (Garthe et al., 2016).

The flight tracks of the gannets shown in Fig. 3 were projected on top of the wind farms that were operational or under construction during the tracking period. Three gannets mostly stayed away from the wind farms and passed around them much more frequently than flying through them (Fig. 10). Wind farms further from Helgoland were not entered, but gannets visited the the areas around them. Focusing on the three wind farms north of Helgoland, five of the 14 gannets tracked in 2015 did not enter them, four only flew into the wind farms once, while four visited them occasionally and one frequently.

## 4 Conclusions and perspectives

Tracking free-living animals such as seabirds can open up new dimensions in biological, ecological, and environmental research (Kays et al., 2015). The latest developments in microelectronics can even provide real-time data transfer through mobile-phone networks (e.g. Gilbert et al., 2016). Information collected by data loggers can be used for various purposes, including applied topics, such as assessing the possible effects of wind farms, as well as fundamental research. In a review of offshore wind-farm studies, Bailey et al. (2014) concluded that traditional visual surveys of birds and mammals from ships and aircraft were unlikely to have enough power to detect changes in behaviour or fine-scale spatial or temporal shifts in distribution, given that observers can only be in one place at a time and can only reliably survey in calm sea conditions during daylight hours. Other techniques such as GPS tracking are thus likely to provide more useful data in many cases (Bailey et al., 2014).

Substantial added-value information can be retrieved by combining geographic-position information with other parameters. For birds feeding under water, pressure sensors are essential to characterise foraging areas, allowing diving activity to be described comprehensively (e.g. Boyd, 1997, Ronconi & Burger, 2011). Pressure sensors and/or high-rate GPS measurements can also be used to estimate flight heights (Corman & Garthe, 2014, Scales et al., 2014). Further detailed behavioural and energetic information can be derived from three-dimensional accelerometer measurements (Gómez Laich et al., 2008, Sakamoto et al., 2009), making this a topical research interest.

To understand the distributions of food-searching seabirds and their variation over time, analysing the birds' habitat choice is an important and promising approach. While some habitat parameters may be collected by the loggers on the birds directly (e.g. sea surface temperatures, Wilson et al., 1995b), a full set of variables can only be derived from a combination of remote-sensing and in-situ measurements. In most studies tracking seabirds, sea surface temperature and chlorophyll have been analysed and compared to bird distributions, often with limited success (e.g. Grémillet et al., 2008). In future activities of our study, we will make use of the project consortium COSYNA (Baschek et al., 2016) that provides comprehensive and relevant information on important habitat variables. We expect that fixed-point measurements may be particular valuable for studying seasonal and/or annual variability of the foraging behaviour and distributions of northern gannets while moving and remote-sensing platforms may be best used to unravel the spatial distribution of the birds at any time. The advantage of COSYNA in this context will be the the variety of measured variables as well as the three-dimensional mesurements so that stratifcation can be assessed which would not be available from remote sensing sources (Baschek et al., 2016). Furthermore, the generation of models may prove particularly valuable for analysing and possibly predicting the distribution of seabirds (Breitbach et al. 2016, Stanev et al. 2016). Finally, information on the marine environment may also be generated through the study of foraging seabirds directly, as their distinct prey-search behaviours may also inform physical oceanographers on the location of physical features, especially small-scale features such as fronts (e.g. Sabarros et al., 2014, Scales et al., 2014).

*Acknowledgements*: This study was funded by the Federal Ministry for Economic Affairs and Energy according to the decision of the German Bundestag (HELBIRD, 0325751). The e-obs loggers used in this study were provided through the COSYNA project led by the Helmholtz Zentrum Geesthacht (HZG). Logistic support on Helgoland was provided by J. Dierschke and the Institute of Avian Research on Helgoland. K. Borkenhagen, R. M. Borrmann, L. Enners, K. Fließbach, N. Guse, J. Jeglinski, K. Lehmann-Muriithi, B. Mendel, S. Müller, G. Schultheiß, H. Schwemmer, S. Vandenabeele, and S. Weiel helped with fieldwork. S. Furness provided linguistic support. All institutional and national guidelines for the handling and equipping of birds were followed. Birds were equipped under a license issued by the Ministry of Energy Transition, Agriculture, Environment and Rural Areas Schleswig-Holstein, Germany (file number: V 312-7224.121-37 (80-6/13)). All animals were handled in strict accordance with good animal practice to minimize handling times and stress.

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

**Table 1. Linear mixed model of dive parameters dive depth and dive duration and their possible explanation by the fixed habitat variables distance to colony, water depth and distance to nearest land. AIC = Akaike's information criterion. LRT = Likelihood ratio test. Significant results in bold.**

| | Dive depth | | | Dive duration | | |
|---|---|---|---|---|---|---|
| | AIC | LRT | p | AIC | LRT | p |
| Full model | 4314.1 | | | 531.2 | | |
| Distance to colony | 4312.4 | 0.265 | 0.606 | 531.9 | 2.747 | 0.097 |
| Water depth | 4312.9 | 0.760 | 0.383 | 537.4 | 8.174 | **0.004** |
| Distance to nearest land | 4315.2 | 3.102 | 0.078 | 530.5 | 1.342 | 0.247 |

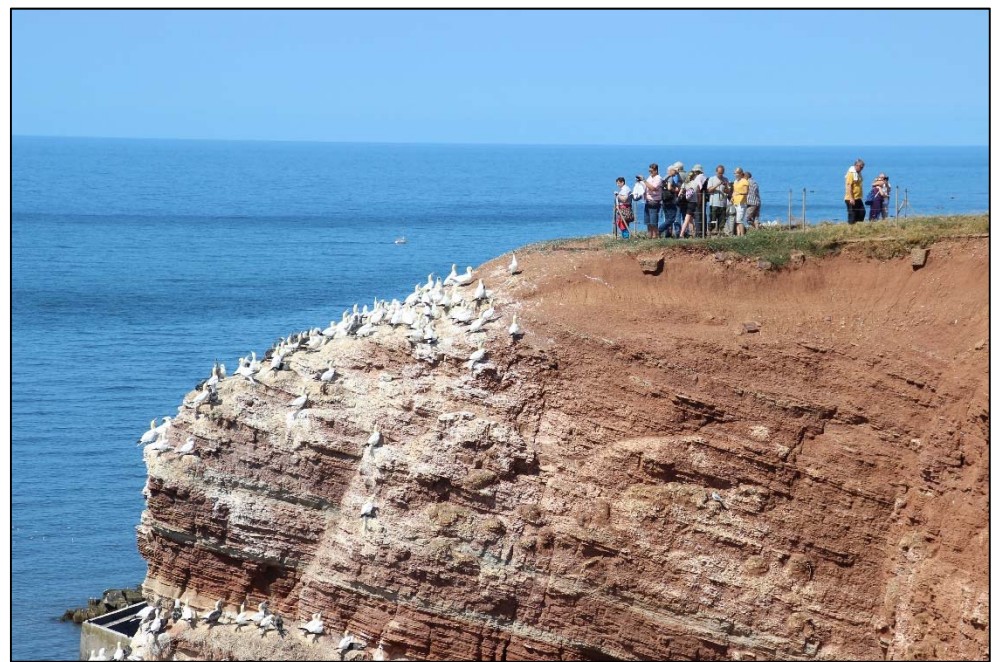

**Figure 1.** Breeding colony of Northern Gannets on the island of Helgoland. Photo: S. Garthe.

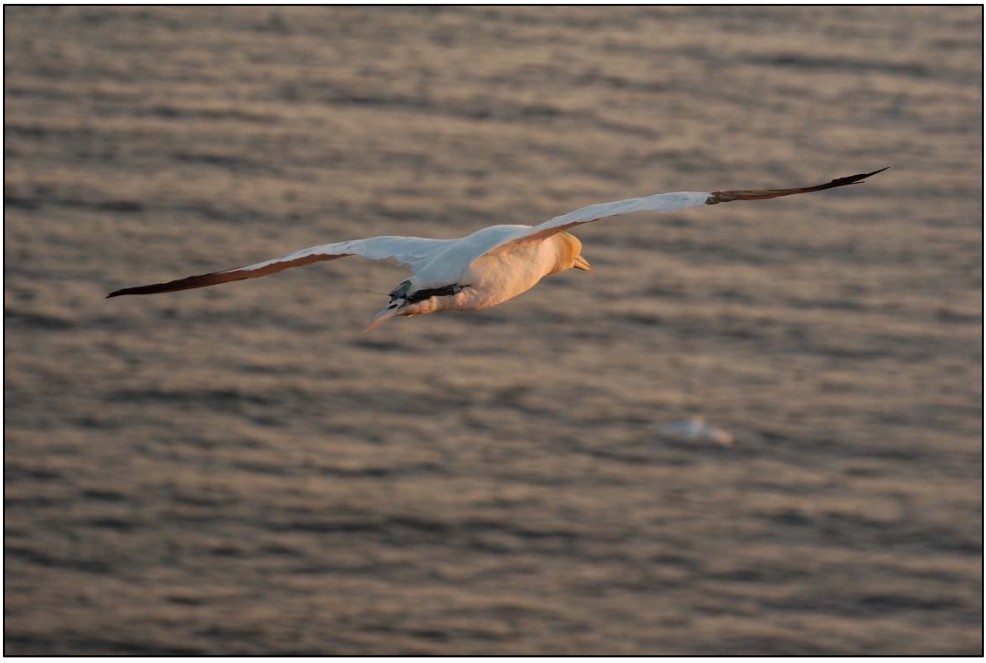

5    **Figure 2.** Flying Northern Gannets with a Bird Solar GPS Logger attached to the tail feathers. Photo: K. Borkenhagen.

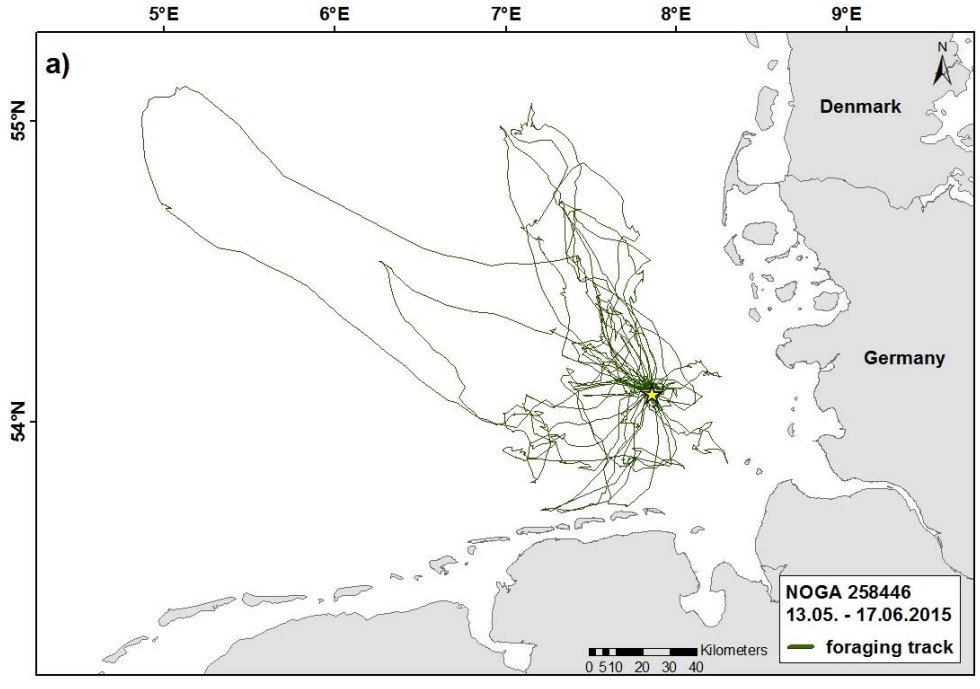

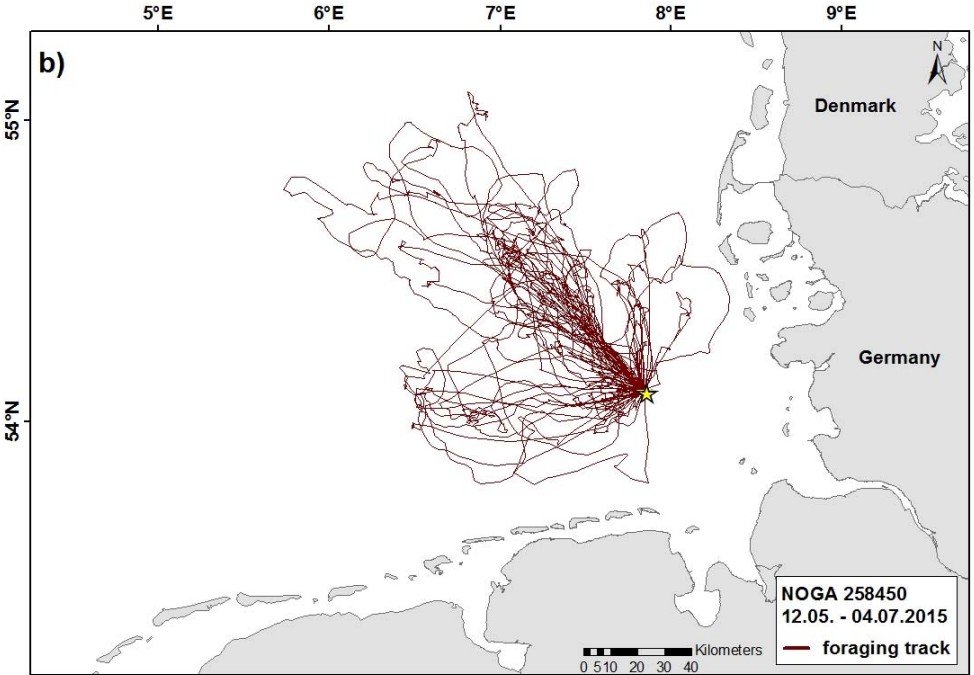

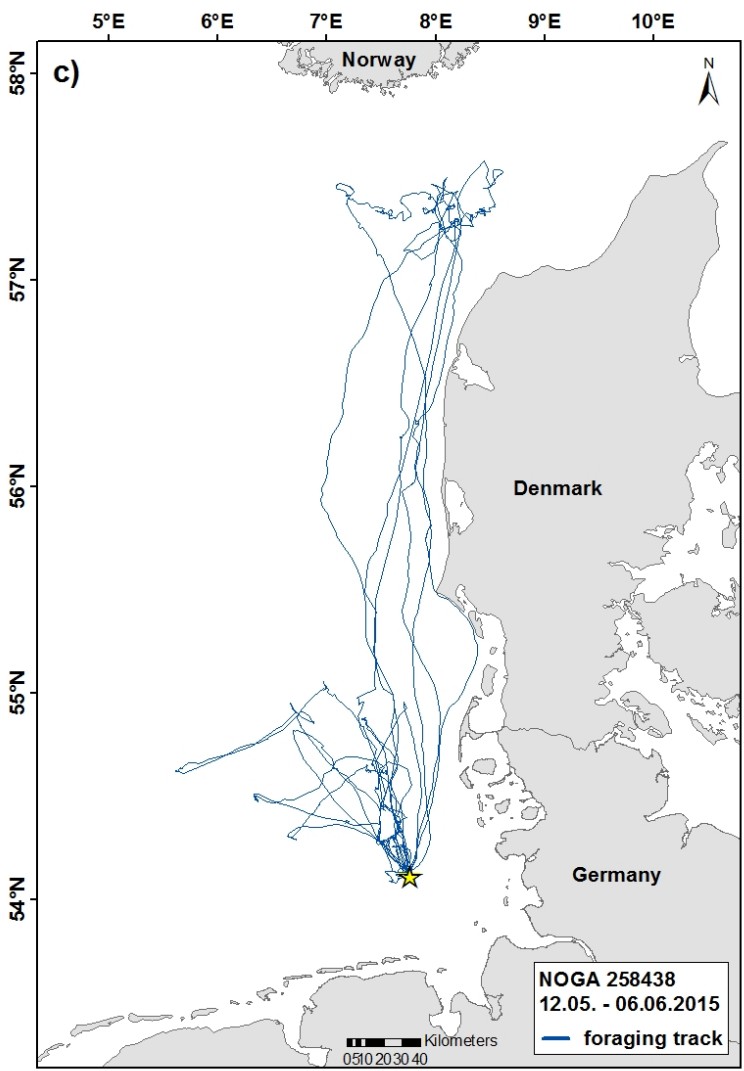

**Figure 3.** Flight patterns of three Northern Gannets (NOGA) breeding on Helgoland in 2015. Birds were tracked over 8 (a), 5 (b), and 3.5 (c) weeks, respectively.

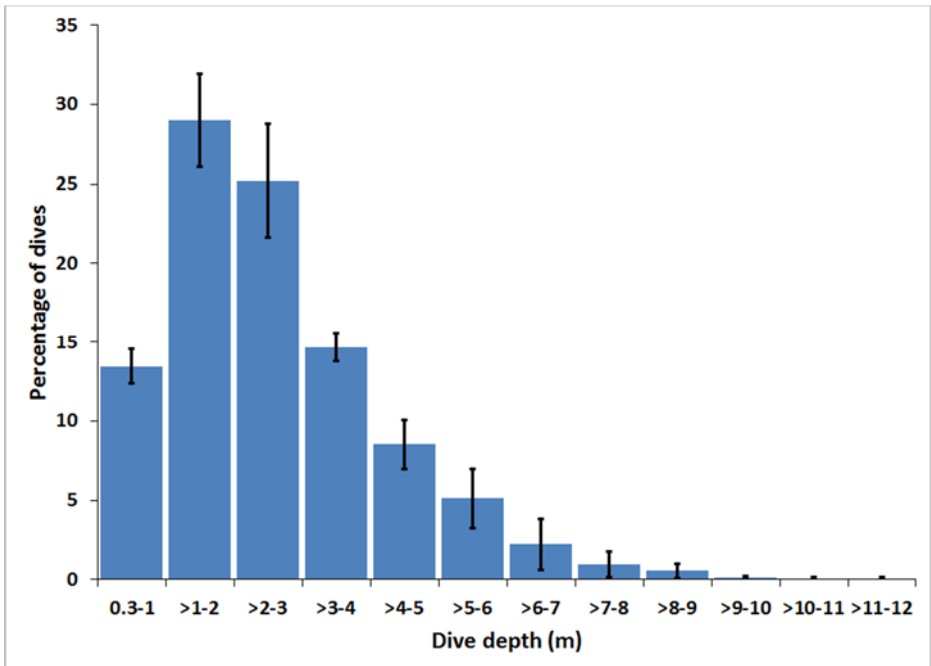

**Figure 4.** Dive depths of Northern Gannets, 2015. Data are based on 2,557 dives recorded from four adults breeding on Helgoland. Vertical bars show values averaged over the four individuals, extended lines show standard errors. Immersions of < 0.3 m were excluded as potentially indicating bathing and other behaviours.

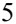

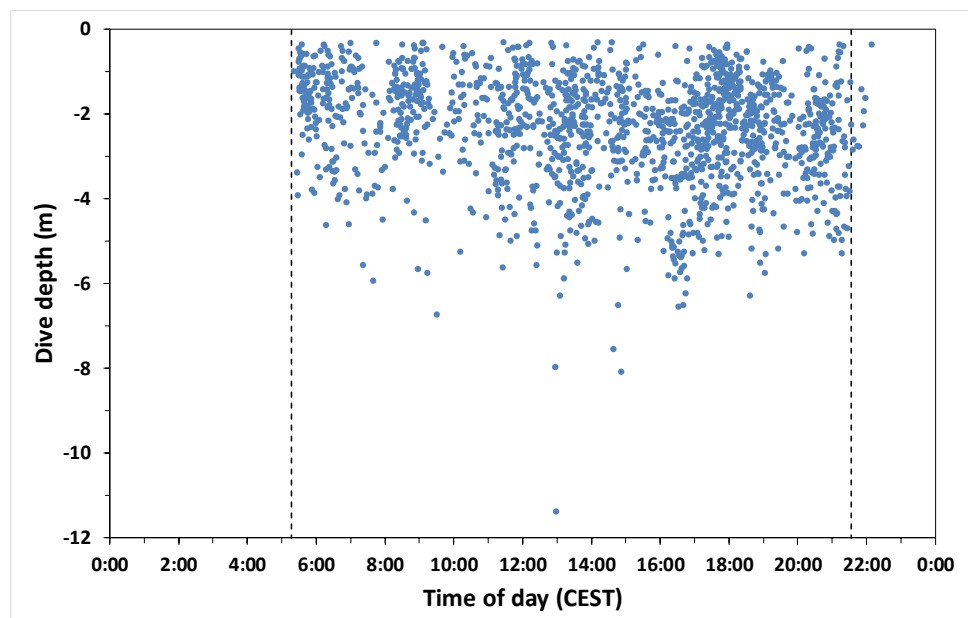

**Figure 5.** Diurnal rhythm in diving activity of Northern Gannets in the German Bight, 2015. Data are for the period 12–31 May 2015 and are based on dive recordings of three adults breeding on Helgoland. Each dot represents one dive, showing the maximum depth during the dive. Vertical dashed lines indicate sunrise and sunset for the median day of the period covered.

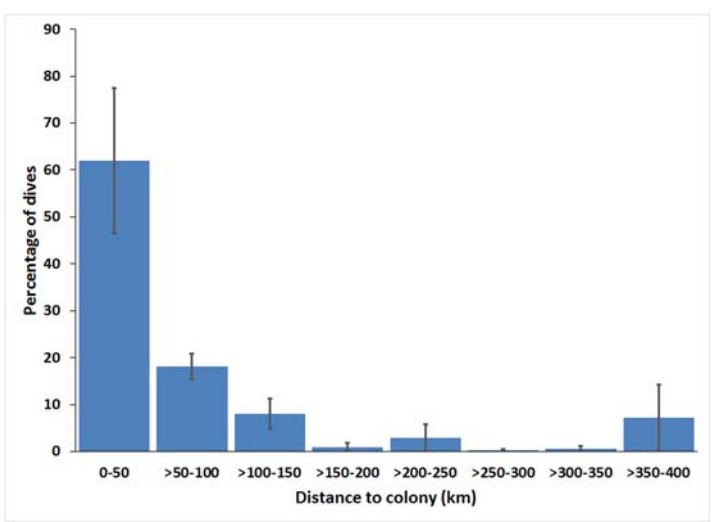

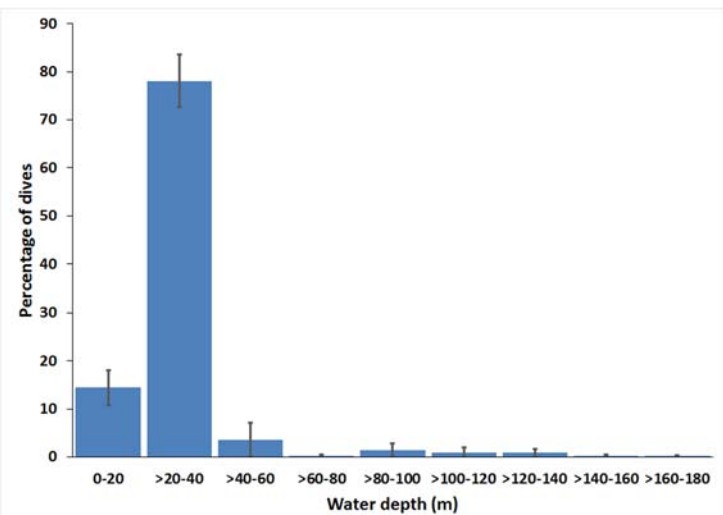

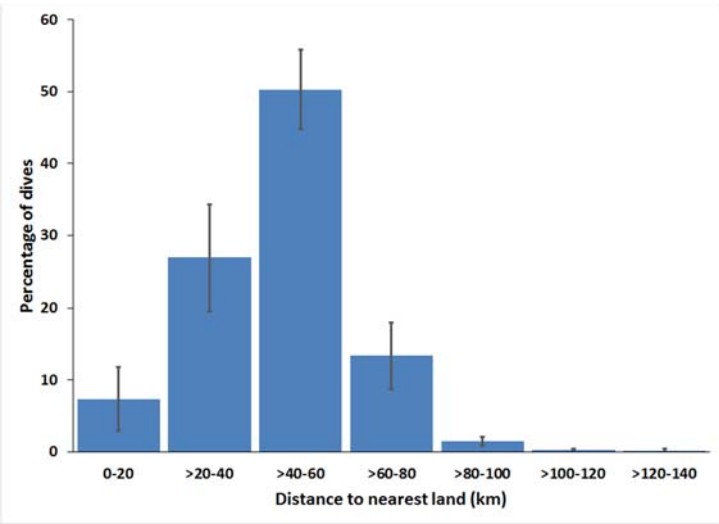

**Figure 6.** Habitat relationships (upper graph: distance to colony, middle graph: distance to nearest land, lower graph: water depth) of diving Northern Gannets, 2015. Data are based on 2,557 dives recorded from four adults breeding on Helgoland. Vertical bars show mean values averaged over the four individuals, extended lines show standard errors.

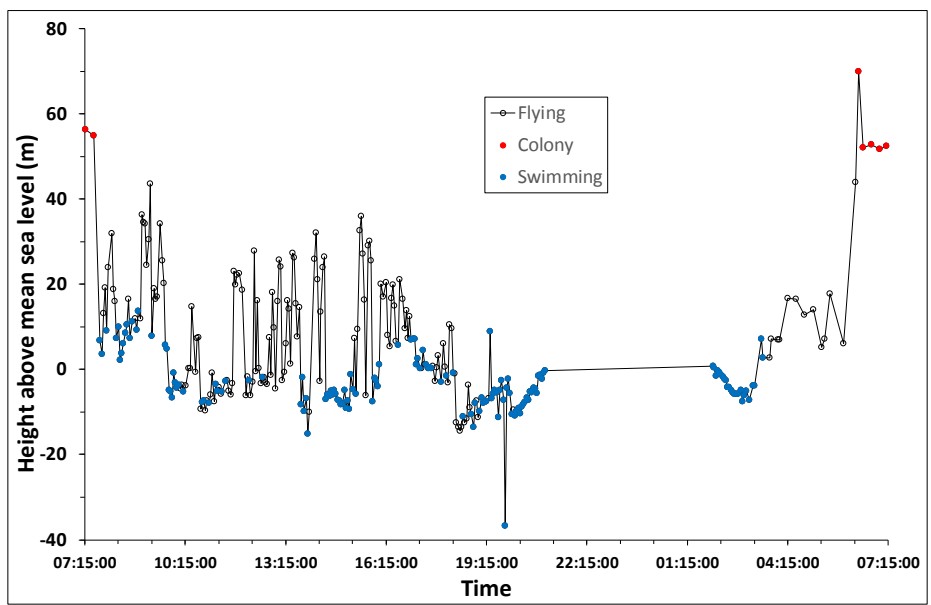

**Figure 7.** Altitude measurements for a Northern Gannet tracked in summer 2015 on Helgoland. Altitude measurements are related to activities 'in colony' (before and after the 22.7 h foraging trip), swimming, and flying. For details see text. Please note that this device was switched-off during the core darkness hours to save energy, and because birds are known to either stay at the nest site or rest at the sea

10 surface during this period (as here).

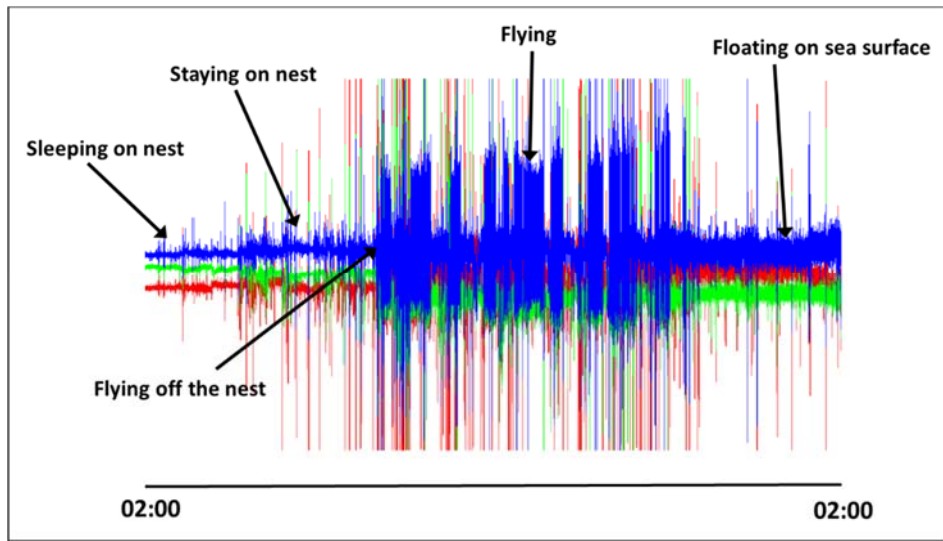

**Figure 8.** Accelerometer measurements for a Northern Gannet breeding on Helgoland. This example shows the values for the three different axes (red = x, green = y, blue = z) over 24 h, from 02:00 to 02:00 CEST on the next day. Different activities are indicated by arrows; higher peak indicate greater movement.

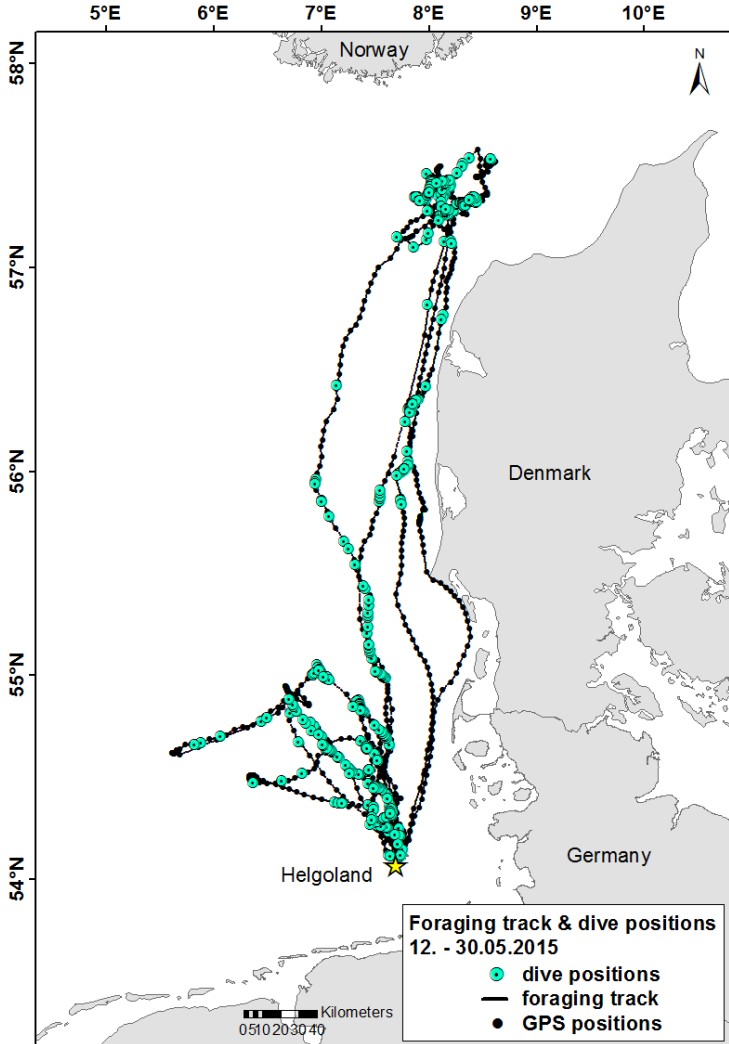

**Figure 9.** Foraging tracks and dive positions for the Northern Gannet shown in Fig. 3c. Please note that the data set is smaller than in Fig. 1c because only the synoptic GPS + pressure data are shown (the memory of the PTD logger was full after about 18 days).

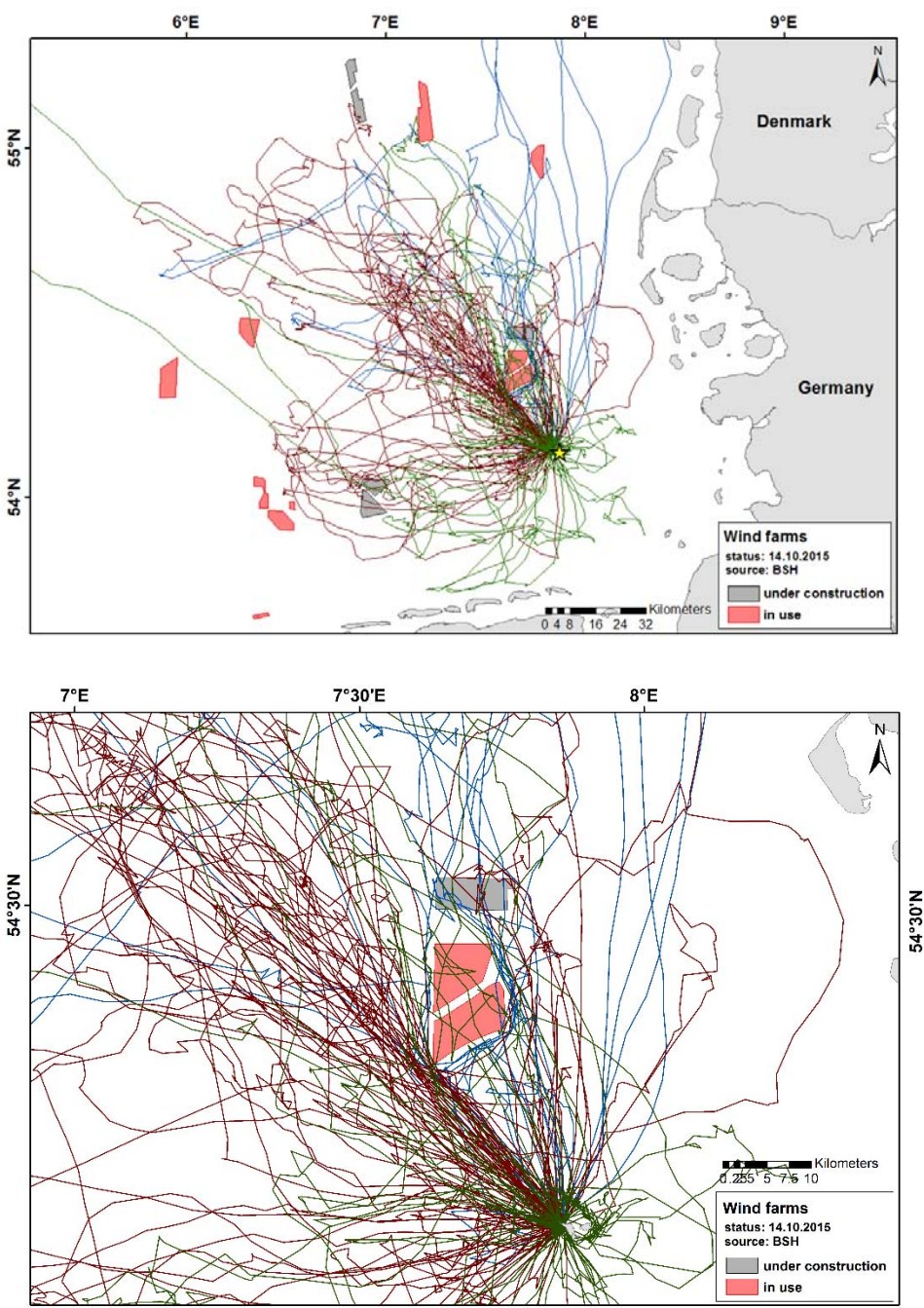

**Figure 10.** Overlap of flight patterns for the three Northern Gannets shown in Fig. 3 with the locations of wind farms in the German Bight. Information on the location and status of wind farms was collated from the Federal Maritime and Hydrographic Agency (BSH, pers. comm.) and from the Global Offshore Wind Farms Database (http://www.4coffshore.com/offshorewind/). The upper graph shows the whole German Bight, the lower graph the area with the three wind farms near Helgoland only.

