# Peer review of "Seabirds as samplers of the marine environment – a case study in Northern Gannets"

_Ocean Science, 2016_

## Referee Comment (RC1) · G. Hunt (Referee) · 3 Aug 2016

Since the pioneering work of Jouventain, Weimerskirch and others, the fitting of seabirds with devices that record or transmit their locations and activities while at sea has provided a wealth of information as to how seabirds relate to oceanographic phenomena at spatial and temporal scales out of reach of either colony-based or ship-based studies. The present paper, by Garthe et al. continues in that vein with the tracking of northern gannets in the German Bight. They report several findings of interest- that some birds repeatedly used the same foraging location, that birds did not focus their foraging on a single location, and that overall, the gannets avoided foraging or passage through a wind farm. The last finding is perhaps good news, as many more wind farms are planned for the German Bight. Their methods seem robust and appropriate, and their analyses are adequate as far as they go.

The above notwithstanding, my feeling is that the authors could have extracted much more information from the data that they amassed. How do their results relate to optimal foraging theory, or ideas about central place foraging? They mention the one bird that flew far to the north, but what about the dispersed nature of the foraging patterns of the other birds? How did the outbound and inbound travel patterns compare? Did a given bird use the same foraging location? Did the flight patterns of high-flying birds differ from those of flight paths at low altitudes? Perhaps the sample sizes necessary for a formal evaluation of these and other questions was not sufficient, or perhaps there are plans for additional papers that will examine these and other questions. How did the foraging locations of individual birds compare with those used by the group as a whole? Clearly there was one outlier, but among the others, what was the relationship of individual variability to the variability of the group as a whole. It would be good to make the most complete use possible of the available data.

Some small stuff:

Page 3, line 4: There are much better references for relating foraging seabirds to prey patches and physical processes: Hunt and Harrison, 1990, MEPS; Hunt et al., 1998, MEPS; Russell et al., 1999, MEPS; Jahncke et al., 2005, Fish Oceanography; Davoren GK, many papers, some with Garthe. Haney did not understand the system in which he was working, and the Decker and hunt paper did not have solid measures of prey distributions.

Page 3, line 22: Your observations were not really experiments in the normal sense of the word.

Page 4, line 6: to what does "respectively refer?

Page 4, line 16: What was the time lag? Minimal is in the eye of the beholder.

Page 4, line 20: It has a mass of 2.3 – 3.6 kg. . ... we do not need the goose-sized. . .

Page 6, line 21: I believe that that should be Figure 7 that you are calling out.

Page 6, line 27: remove "actively". It adds no information.

Page 7, line 8: I believe that that should be Fig. 3 that is being called out.

Figures 1 and 2: Both nice, but not necessary.

Figure 9: Maybe add an insert to show more clearly what happens around gthe wind farm closest to the colony?
* * *

---

## Referee Comment (RC2) · M. Frederiksen (Referee) · 3 Oct 2016

This brief manuscript presents some results from one season of tracking of Northern Gannets at a German colony. These results are interesting and nicely presented. However, I'm not sure I fully understand exactly what the authors are trying to do here – what is the aim of the manuscript, and which questions are the authors trying to answer? The manuscript seems to simply be a case study of what can be learned about seabird foraging behavior through biologging. I guess the lack of a broader context and clear aims is related to the fact that the manuscript forms part of a special issue, and that the authors are trying to show how seabirds can play a part in an ocean observation network. However, in my opinion the manuscript should also be accessible and interesting to readers who're unfamiliar with COSYNA, and this will require a bit more context, more clearly defined aims, and a discussion of the pros and cons of the

chosen approach in terms of reaching those aims.

It is difficult to come up with more specific comments. The examples chosen are neatly presented, and the methodology appears to be up to date, although perhaps not innovative. The authors could without doubt have chosen to present other interesting results, I imagine some of these will be presented in other contexts. The results on dive behavior are based on a minuscule sample size (4 individuals), and this should be kept in mind when interpreting the results (e.g. differences in dive depth relative to other studies).

Minor comments: - P- 2, l. 15: 'have a strong influence on . . .' or similar. - P. 3, l. 28: perhaps more informative to say 'central' tail feathers. - P. 6, l. 21: the reference here should be to Fig. 7. - Fig. 7: please add at least a time axis. I guess the acceleration axis is difficult to label informatively.

---

## Editor Comment (EC1) · H. Brix (Editor) · 4 Nov 2016

In addition to the very helpful comments by the two reviewers I would like to encourage the authors to see to the following points:

- the quantitative results of this manuscript should be backed up by statistical analyses of the data used. So far it reads more like a collection of case studies. It would also be nice to see statistically based error estimates where feasible.

- the manuscript does not reference "outside" data, nor does it provide a comparison with other studies either on the same species or location.

- no use is made of the data sets available in COSYNA or, for instance, through the Helgoland Roads time-series station. As this paper is part of the COSYNA special issue

the link to and synergies with the other components of COSYNA should be discussed, for the past as well as with regard to a future potential for cooperation.

Some minor comments are to be found in the annotated manuscript.

Please also note the supplement to this comment:
http://www.ocean-sci-discuss.net/os-2016-22/os-2016-22-EC1-supplement.pdf
* * *
[Figure]

**Supplement:**

[revised manuscript text omitted]

---

## Author Comment (AC1) · 16 Dec 2016

AUTHORS: Thank you very much for your time and your valuable suggestions that helped us improving our manuscript.

REVIEWER: Since the pioneering work of Jouventain, Weimerskirch and others, the fitting of seabirds with devices that record or transmit their locations and activities while at sea has provided a wealth of information as to how seabirds relate to oceanographic phenomena at spatial and temporal scales out of reach of either colony-based or ship-based studies. The present paper, by Garthe et al. continues in that vein with the tracking of northern gannets in the German Bight. They report several findings of interest- that some birds repeatedly used the same foraging location, that birds did not focus their foraging on a single location, and that overall, the gannets avoided foraging or passage through a wind farm. The last finding is perhaps good news, as many more wind farms are planned for the German Bight. Their methods seem robust and appropriate, and their analyses are adequate as far as they go.

AUTHORS: Thank you.

REVIEWER: The above notwithstanding, my feeling is that the authors could have extracted much more information from the data that they amassed. How do their results relate to optimal foraging theory, or ideas about central place foraging? They mention the one bird that flew far to the north, but what about the dispersed nature of the foraging patterns of the other birds? How did the outbound and inbound travel patterns compare? Did a given bird use the same foraging location? Did the flight patterns of high-flying birds differ from those of flight paths at low altitudes? Perhaps the sample sizes necessary for a formal evaluation of these and other questions was not sufficient, or perhaps there are plans for additional papers that will examine these and other questions.

AUTHORS: We understand this comment very well. While one paper focusing on gannets and wind farms near Helgoland – using data from 2014 – has just been published (and is referred to, now), several other papers focusing on specific aspects are under preparation. To nonetheless address your comment, we have created a whole new section, chapter 3.3, habitat analyses, for which we have comprehensively analysed the dive data shown in chapter 3.2 in relation to habitat variables. This does not only include means + standard errors for the data, but also a mixed model analysing habitat variables. Apart from the text addressing this topic, we have also created a new figure (containing three maps) and a table.

REVIEWER: How did the foraging locations of individual birds compare with those used by the group as a whole? Clearly there was one outlier, but among the others, what was the relationship of individual variability to the variability of the group as a whole. It would be good to make the most complete use possible of the available data.

AUTHORS: A separate paper will deal with this issue based on data from 2015 and 2016. This would be far outside the scope for this manuscript but is certainly a very interesting question.

REVIEWER: Some small stuff: Page 3, line 4: There are much better references for relating foraging seabirds to prey patches and physical processes: Hunt and Harrison, 1990, MEPS; Hunt et al., 1998, MEPS; Russell et al., 1999, MEPS; Jahncke et al., 2005, Fish Oceanography; Davoren GK, many papers, some with Garthe. Haney did not understand the system in which he was working, and the Decker and hunt paper did not have solid measures of prey distributions.

AUTHORS: We have exchanged the references as suggested.

REVIEWER: Page 3, line 22: Your observations were not really experiments in the normal sense of the word.

AUTHORS: Renamed to 'field work'.

REVIEWER: Page 4, line 6: to what does "respectively refer?

AUTHORS: Reworded.

REVIEWER: Page 4, line 16: What was the time lag? Minimal is in the eye of the beholder.

AUTHORS: Details have been provided now including a reference.

REVIEWER: Page 4, line 20: It has a mass of 2.3 – 3.6 kg: : :.. we do not need the goose-sized: : :

AUTHORS: Sure; removed.

REVIEWER: Page 6, line 21: I believe that that should be Figure 7 that you are calling out.

AUTHORS: Yes; corrected.

REVIEWER: Page 6, line 27: remove "actively". It adds no information.

AUTHORS: Removed.

REVIEWER: Page 7, line 8: I believe that that should be Fig. 3 that is being called out.

AUTHORS: Yes; corrected

REVIEWER: Figures 1 and 2: Both nice, but not necessary.

AUTHORS: In the context of a primarily hydrographic-oriented research program I would suggest to keep these photos to illustrate the study objects.

REVIEWER: Figure 9: Maybe add an insert to show more clearly what happens around the wind farm closest to the colony?

AUTHORS: Sure; we provide two maps now, the first focusing on the whole German Bight, the second on the three wind farms north of Helgoland only.

---

## Author Comment (AC2) · 16 Dec 2016

AUTHORS: Thank you very much for your time and your valuable suggestions that helped us improving our manuscript.

REVIEWER: This brief manuscript presents some results from one season of tracking of Northern Gannets at a German colony. These results are interesting and nicely presented.

AUTHORS: Thank you.

REVIEWER: However, I'm not sure I fully understand exactly what the authors are trying to do here – what is the aim of the manuscript, and which questions are the authors trying to answer? The manuscript seems to simply be a case study of what

can be learned about seabird foraging behavior through biologging. I guess the lack of a broader context and clear aims is related to the fact that the manuscript forms part of a special issue, and that the authors are trying to show how seabirds can play a part in an ocean observation network. However, in my opinion the manuscript should also be accessible and interesting to readers who're unfamiliar with COSYNA, and this will require a bit more context, more clearly defined aims, and a discussion of the pros and cons of the chosen approach in terms of reaching those aims.

AUTHORS: This is a good point that we fully understand. Therefore, a link to other COSYNA components has been established so that this manuscript is better integrated into this special issue.

REVIEWER: It is difficult to come up with more specific comments. The examples chosen are neatly presented, and the methodology appears to be up to date, although perhaps not innovative. The authors could without doubt have chosen to present other interesting results, I imagine some of these will be presented in other contexts.

AUTHORS: We understand this comment very well. While one paper focusing on gannets and wind farms near Helgoland – using data from 2014 – has just been published (and is referred to, now), several other papers focusing on specific aspects are under preparation. To nonetheless address your comment, we have created a whole new section, chapter 3.3, habitat analyses, for which we have comprehensively analysed the dive data shown in chapter 3.2 in relation to habitat variables. This does not only include means + standard errors for the data, but also a mixed model analysing habitat variables. Apart from the text addressing this topic, we have also created a new figure (containing three maps) and a table.

REVIEWER: The results on dive behavior are based on a minuscule sample size (4 individuals), and this should be kept in mind when interpreting the results (e.g. differences in dive depth relative to other studies).

AUTHORS: We fully agree that data have to be interpreted with caution. However, the

consistently shallow dives are different from all the other gannet diving studies that we carried out in eastern Canada, even if subsampling small data sets. This has been written out explicitly in the manuscript.

REVIEWER: Minor comments: - P- 2, l. 15: 'have a strong influence on : : :' or similar.

AUTHORS: Reworded.

REVIEWER: - P. 3, l. 28: perhaps more informative to say 'central' tail feathers.

AUTHORS: Corrected.

REVIEWER: - P. 6, l. 21: the reference here should be to Fig. 7.

AUTHORS: Yes; corrected.

REVIEWER: - Fig. 7: please add at least a time axis. I guess the acceleration axis is difficult to label informatively.

AUTHORS: Yes. The time axis has been added.

---

## Author Comment (AC3) · 16 Dec 2016

AUTHORS: Thank you very much for your time and your valuable suggestions that helped us improving our manuscript.

EDITOR: In addition to the very helpful comments by the two reviewers I would like to encourage the authors to see to the following points: - the quantitative results of this manuscript should be backed up by statistical analyses of the data used. So far it reads more like a collection of case studies. It would also be nice to see statistically based error estimates where feasible.

AUTHORS: We understand this comment very well. While one paper focusing on gannets and wind farms near Helgoland – using data from 2014 – has just been published (and is referred to, now), several other papers focusing on specific aspects are under

preparation. To nonetheless address your comment, we have created a whole new section, chapter 3.3, habitat analyses, for which we have comprehensively analysed the dive data shown in chapter 3.2 in relation to habitat variables. This does not only include means + standard errors for the data, but also a mixed model analysing habitat variables. Apart from the text addressing this topic, we have also created a new figure (containing three maps) and a table.

EDITOR: - the manuscript does not reference "outside" data, nor does it provide a comparison with other studies either on the same species or location.

AUTHORS: We have inserted several references to studies on this species from other locations, enabling some comparisons. With regard to Helgoland, we have not been able to study any other species on Helgoland with the same technology until spring 2016. Now, we have started to investigate two further species, but data have not been published yet (nor will be within the next few months).

EDITOR: - no use is made of the data sets available in COSYNA or, for instance, through the Helgoland Roads time-series station. As this paper is part of the COSYNA special issue the link to and synergies with the other components of COSYNA should be discussed, for the past as well as with regard to a future potential for cooperation.

AUTHORS: A time-series analyses is not yet possible on our data set as we could not track gannets in sufficient years. However, a link to other COSYNA components has been established so that this manuscript is better integrated into this special íssue.

EDITOR: Some minor comments are to be found in the annotated manuscript.

AUTHORS: We have incorporated your comments into the text.

---

## Editor Decision (ED1)

The manuscript has improved substantially and most of the reviewers' questions and concerns have been addressed. There are a few points the authors might look into before this manuscript is ready for publication:

First line of abstract: typo "requires"

Page 3, line 9: the reference to Fig. 2 might be placed better at the end of the preceding sentence.

Page 5, last paragraph: Please revise your description of the linear mixed model. This paragraph stands somewhat detached from the rest of the manuscript and is hard to understand not knowing the program used. The authors should concentrate their discussion here on a clear explanation of the method and less on the technical details such as names of functions that are not instructive to the reader. What does this function drop1 do, what is the test about? Testing and omitting of terms in the model cannot be understood without further information on the tests.

Page 6, lines 24-30: While the method is clear, I miss some explanation on the relevance of these findings. They are only an example from one animal. Do other animals behave in a similar way? Or has this only been measured for one animal? What are the conclusions here? Birds are more active when they fly, than when sitting on their nests and they take multiple breaks? Please elaborate.

Page 8, line 7: What is meant by "In our case study…"? Does that refer to future work? Please elaborate.

After addressing these points the manuscript should be ready for publication.

---

## Author Response (AR2)

**EC1:**

EDITOR: The manuscript has improved substantially and most of the reviewers' questions and concerns have been addressed. There are a few points the authors might look into before this manuscript is ready for publication:

AUTHORS: Thank you for your efforts and your positive review.

EDITOR: First line of abstract: typo "requires"

AUTHORS: Corrected.

EDITOR: Page 3, line 9: the reference to Fig. 2 might be placed better at the end of the preceding sentence.

AUTHORS: Changed as suggested.

EDITOR: Page 5, last paragraph: Please revise your description of the linear mixed model. This paragraph stands somewhat detached from the rest of the manuscript and is hard to understand not knowing the program used. The authors should concentrate their discussion here on a clear explanation of the method and less on the technical details such as names of functions that are not instructive to the reader. What does this function drop1 do, what is the test about? Testing and omitting of terms in the model cannot be understood without further information on the tests.

AUTHORS: We tried to describe what we did to enable the reader to adopt such a test. To address your comments, we have re-worded the whole paragraph.

EDITOR: Page 6, lines 24-30: While the method is clear, I miss some explanation on the relevance of these findings. They are only an example from one animal. Do other animals behave in a similar way? Or has this only been measured for one animal? What are the conclusions here? Birds are more active when they fly, than when sitting on their nests and they take multiple breaks? Please elaborate.

AUTHORS: These analyses still wait to be done. The decription shows the potential of the method. I have added more background on the relevance of such information. Good point, thank you.

EDITOR: Page 8, line 7: What is meant by "In our case study…"? Does that refer to future work? Please elaborate.

AUTHORS: Clarified. I could come up with more possible options what to do but I do not like very much to announce 'big plans' – I'd rather try to start doing it. But of course, if needed, I'd be happy to extend the following sentences of the paragraph.

EDITOR: After addressing these points the manuscript should be ready for publication.

AUTHORS: Thank you, this sounds very good.

[revised manuscript text omitted]

---

## Editor Decision (ED2)

I thank the authors for the improvements in version 4 of their manuscript. Unfortunately, I do still see some adjustments needed in regard to two of the points raised in my comments on version 3.

Page 5, last paragraph:

AUTHORS: We tried to describe what we did to enable the reader to adopt such a test. To address your comments, we have re-worded the whole paragraph.

I still see some need for additional clarification of the method. Especially the part on model selection in the last line of page 5 remains unclear. If "a Gaussian linear mixed model" is used, where does the model selection come in there ("a" model implies for me there is only one, so where is the choice made? Or is "model" used here in a different meaning?).

In the first lines of page 6 I still do not understand what the whole concept of terms entering the model is about. Why does sequence matter? My guess would be that it is about linear terms explaining a certain part of the probability but the reader is still left guessing.

Page 6, lines 24-30:

AUTHORS: These analyses still wait to be done. The decription shows the potential of the method. I have added more background on the relevance of such information. Good point, thank you.

Please add a sentence clarifying that the analysis will be done for more birds in a separate study.

After addressing these points the manuscript should be ready for publication.

---

## Author Response (AR3)

**EC1:**

EDITOR: I thank the authors for the improvements in version 4 of their manuscript. Unfortunately, I do still see some adjustments needed in regard to two of the points raised in my comments on version 3.

AUTHORS: OK, we have worked on this, see below. Thanks for your comments!

EDITOR: Page 5, last paragraph: I still see some need for additional clarification of the method. Especially the part on model selection in the last line of page 5 remains unclear. If "a Gaussian linear mixed model" is used, where does the model selection come in there ("a" model implies for me there is only one, so where is the choice made? Or is "model" used here in a different meaning?). In the first lines of page 6 I still do not understand what the whole concept of terms entering the model is about. Why does sequence matter? My guess would be that it is about linear terms explaining a certain part of the probability but the reader is still left guessing.

AUTHORS: OK, we have re-written part of this paragraph. In the journals we usually publish in, the explanations are relatively short, so as we described them in the first revision. But across discplines, this may not be as straightforward, therefore we have added again more background. Actually, we ran two models, one each for dive depth and dive duration, respectively. To get the relevant p-values for the lmer-output/the output of a linear mixed model using 'lmer' either the functions 'anova' or 'drop1' are commonly used. 'Anova' returns sequential frequentist F-tests, i.e. the order of the predictors is relevant, while 'drop1' returns marginal frequentist F-tests. The latter tests for the same term whether it explains the remaining variance significantly after having corrected for all the other terms in the model (Korner-Nievergelt et al., 2015). That is why we chose to use drop1 instead of anova for testing the influence of each predictor on the respecting response variable.

EDITOR: Page 6, lines 24-30: Please add a sentence clarifying that the analysis will be done for more birds in a separate study.

AUTHORS: Done.